# Development of a Solar Tracking-Based Movable Louver System to Save Lighting Energy and Create a Comfortable Light Environment

**Su-yeon Jung** [1], **Sowon Han** [2], **Min-Goo Lee** [3] **and Heangwoo Lee** [4],*

1   Major of Interior Design, College of Design, Sangmyung University, Cheonan-si 31066, Republic of Korea
2   Major in Space Design, College of Design, Sangmyung University, Cheonan-si 31066, Republic of Korea
3   Korea Electronics Technology Institute, Seongnam-si 13509, Republic of Korea
4   College of Design, Sangmyung University, Cheonan-si 31066, Republic of Korea
*   Correspondence: 2hw@smu.ac.kr; Tel.: +82-41-550-5215

**Abstract:** Louvers are among the technical components considered for improving energy performance in buildings, and there has been increased interest in adapting the louver system. However, most previous studies have focused on their performance evaluation based on the width and angle of the slats, which allow for limited improvement in their efficiency. This study suggests a solar tracking-based movable louver (STML) system and examines the efficacy using a full-scale test bed. To do so, we developed a full-scale test bed and estimated the energy reduction and improvement of indoor uniformity of different types of STML systems, including vertical, horizontal, eggcrate, and hybrid. The main findings are as follows: (1) The proposed STML is a hybrid louver with four movable shafts due to its structural characteristics. The shading area is increased sequentially by controlling the length of the movable shaft adjacent to the sun through solar tracking. (2) Compared with conventional vertical and horizontal louvers, the STML can improve indoor uniformity by 5.0% and 13.9%, respectively. Unlike conventional louvers, the STML awnings are installed at the end of the daylighting window, reducing window view obstruction and creating a more pleasant indoor visual environment. (3) Compared with conventional louvers, the STML can reduce lighting and heating/cooling energy by 35.7–49.7%. These findings prove the effectiveness of the proposed system.

**Keywords:** louver; sun tracking; energy saving; daylighting environment; performance evaluation

## 1. Introduction

According to the "2021 Global Status Report for Buildings and Construction" by the Global Alliance of Buildings and Construction (GABC), buildings accounted for 36% of global energy demand and 37% of energy-related $CO_2$ emissions in 2020, which is higher than those of the transport and industrial sectors [1]. In this respect, energy consumption and $CO_2$ emissions in the building sector will remain on the rise, necessitating the need for research and development (R&D) to solve these problems [2,3]. Building envelope performance is a critical factor in determining how energy-efficient a building is [4–7], so research to enhance this performance is crucial for reducing building energy use. The main functions of the building envelope are not daylighting and shading as these factors vary among functional requirements. Therefore, numerous studies have been conducted on related technical elements, including blinds [8–10], louvers [11,12], awnings [13], and light shelves [14–16]. Among these elements, a louver is a type of horizontal or vertical awning. Despite its simple structure, it is widely applied to buildings because of its high-performance efficiency [17,18]. However, previous studies on louvers mainly focused on determining the optimal specifications, such as the angle and width of slats to respond to the external environment [19,20], implying that they cannot be an effective alternative for saving building energy and creating a comfortable indoor environment. Recently, many

studies have combined shading and daylighting technologies with information technologies to maximize building energy savings and create a comfortable indoor environment [21]. In this respect, there is an increasing demand for advanced research and technical review of louvers to improve their shading and daylighting efficiency.

Therefore, this study proposes an optimal type of louver and its control method for use in buildings and validates its performance using a test bed.

### 1.1. Concepts and Technologies Related to Louvers

A daylighting system refers to various systems in buildings that admit natural light into rooms where natural light is difficult to introduce. These systems help create a pleasant visual environment while also conserving energy [22]. An awning system is used to block or regulate the amount of natural light that enters from the outside to reduce the cooling load and discomfort glare of indoor spaces, particularly during the summer. However, these awning systems tend to increase the lighting energy consumption of indoor spaces by reducing the amount of natural light entering the room, thereby increasing the lighting energy consumption of buildings [23,24]. Typical examples of such daylighting and shading systems include louvers, awnings, blinds, and light shelves. Among these systems, louvers are one of the most commonly used shading systems to partially or completely block sunlight. However, as shown in Figure 1, louvers are used not only for shading but also serve as daylighting systems by reflecting natural light using highly reflective materials. As shown in Figure 2, there are various types of louvers, including vertical, horizontal, and eggcrate louvers. Vertical louvers are effective when installed in the east–west direction of buildings, where they function as windshields to help insulate the glass during winter [25]. Horizontal louvers are generally installed in the south-facing direction to block natural light from the high solar altitude during summer. Additionally, by enabling sunlight to penetrate the building, they help buildings save energy during the winter. Because of their distinct characteristics, these vertical and horizontal louvers are applied in different environments. In this respect, hybrid louvers were introduced recently, but they have limitations in responding to various external environments because they can only be combined with the existing mechanisms of vertical and horizontal louvers. Furthermore, applying louvers to windows may obstruct the view from the windows [26].

Table 1 shows the results of previous studies on louvers [27–35], most of which focused on evaluating the performance of horizontal and vertical-type louvers. Particularly, they focused on the performance of specific slat angles rather than the operation of the louvers. Additionally, there is a significant lack of research on hybrid or complex louvers, with some studies on hybrid louvers [35] combining only horizontal and vertical louvers with no operation controls. In this respect, the STML proposed in this study is more distinct and comprehensive than in previous studies.

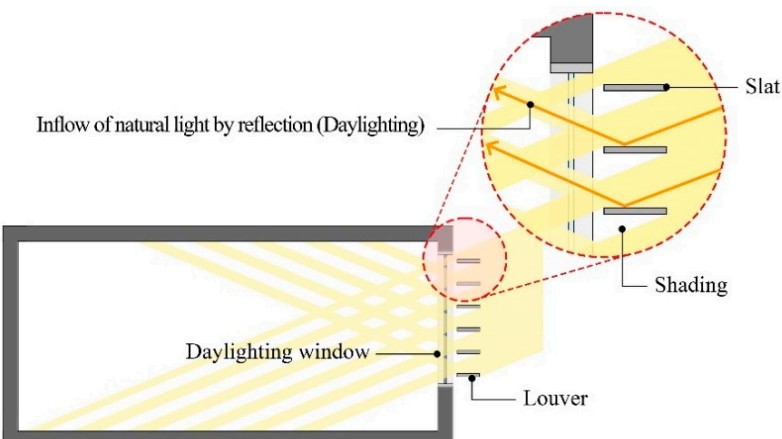

**Figure 1.** Shading and daylighting by a louver.

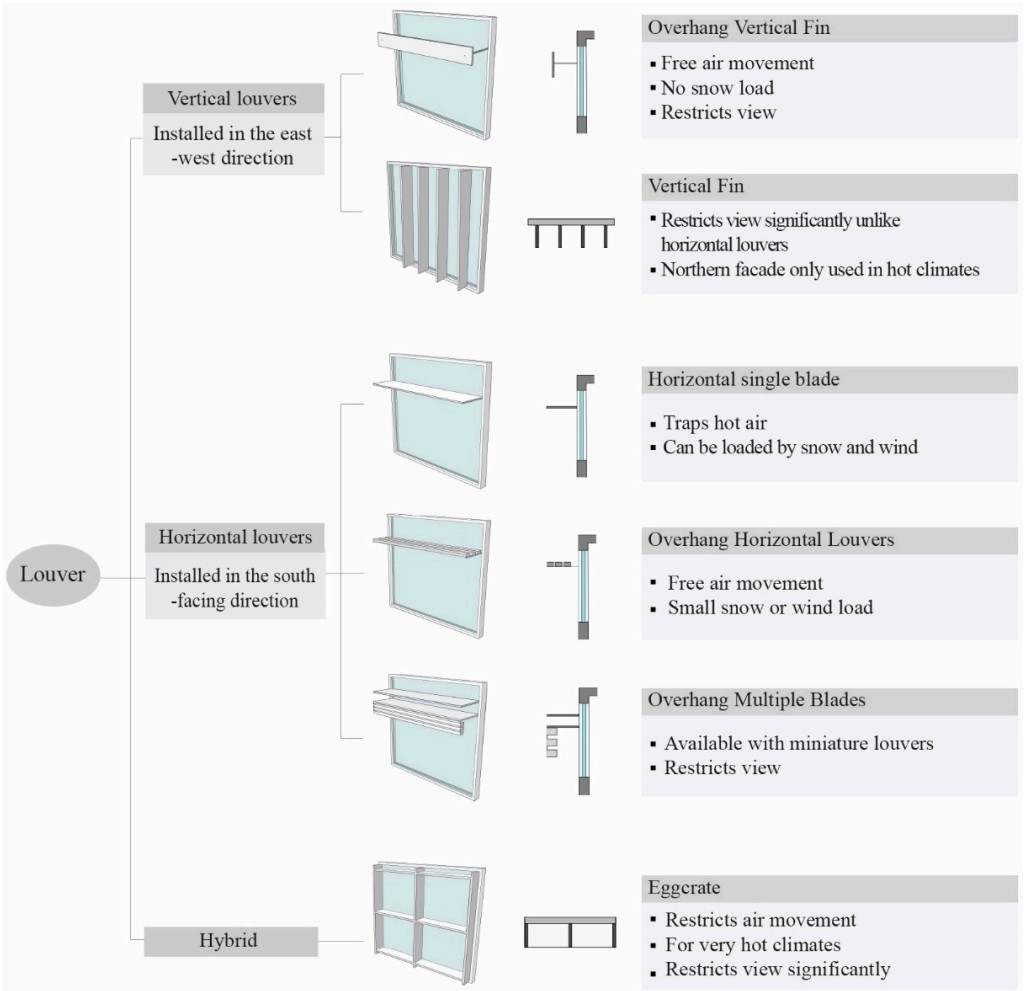

**Figure 2.** Types and characteristics of louvers.

**Table 1.** Consideration of previous work on louver shading systems.

| Author (Year) | Purpose | Louver Type | Operation of Louver Slats |
|---|---|---|---|
| Uribe et al. (2019) [27] | Derive control strategies for perforated curved louvers for visual comfort and energy savings in office buildings | Horizontal | Fixed at 30°, 45°, 60° |
| Ahmed A et al. (2009) [28] | Performance evaluation of louvers according to ceiling geometry | Horizontal | Fixed at 0° |
| Ana I and Armando C (2010) [29] | Performance evaluation of louver shading devices by region and façade | Vertical, Horizontal | Fixed at 10°, 20°, 30°, 40°, 50°, 60°, 70°, 80°, 90° |
| Hammad and Abu-Hijleh (2010) [30] | Analysis of energy performance using exterior louvers in office buildings | Vertical, Horizontal | Fixed at −80°, −60°, −40°, −20°, 0°, 20°, 40°, 60°, 80° |
| Hernández et al. (2017) [31] | Effects of louvers shading devices on visual comfort and energy savings in an office building | Vertical, Horizontal | Vertical fixed at 0°, 30°, 60°, −30°, −60° /horizontal fixed at 0° |
| Hussain H and Amneh H (2010) [32] | Assessment of daylighting quality and energy-saving performance of horizontal and vertical shading devices | Vertical, Horizontal | Fixed at 0°, 45° |
| Datta (2001) [33] | Thermal performance analysis of horizontal louver devices by TRNSYS simulation | Horizontal | Fixed at 0°, 30°, 45°, 60° |
| Pourshab et al. (2020) [34] | Airflow analysis in an office building with louvers and double-glazed façades | Vertical, Horizontal | Fixed at 0° |
| Horner et al. (2014) [35] | To propose a design approach and evaluate the performance of site-specific louvered shells | Eggcrate | Fixed |

### 1.2. Solar Tracking Technology

Solar tracking is a technology that is used to orient a payload toward the sun as it moves across the sky, and it plays a significant role in various solar energy applications [36,37]. As shown in Table 2, solar tracking systems are generally implemented by calculating the position of the sun or using optical sensors [38]. The details are as follows. First, solar tracking applications that calculate the sun's position necessitate high precision. The sun's position is calculated based on the latitude, altitude, time of sunrise and sunset, and solar altitude of the target area of the sun to be tracked. However, the disadvantage of this method is that it cannot reflect weather conditions and shadows cast by adjacent buildings [39,40]. Solar tracking by optical sensors tracks the brightest spot in the sky. Although this method is imprecise, it allows the sun to be tracked while considering atmospheric conditions [41].

**Table 2.** Solar tracking technologies.

| Method | Characteristics |
| --- | --- |
| Calculating the sun's position | · Calculating the sun's position considering the solar altitude and the time of sunrise and sunset<br>· Using a predetermined value, the error rate is low by tracking the sun regardless of the weather |
| Using illuminance sensors | · Solar tracking is possible without any restrictions on geographical locations<br>· Requires a separate optical sensor, and the error rate is high because the operation of the optical sensor is affected by weather conditions such as cloudiness or diffused radiation |

## 2. Method

### 2.1. Proposal for a Solar Tracking-Based Movable Louver (STML) System

As previously stated, most previous studies on louvers analyzed their performance by only adjusting the slat angles of horizontal and vertical louvers. However, horizontal and vertical louvers each have their respective advantages and disadvantages; hence, there is a limit to simultaneously maximizing shading and daylighting efficiency based on real-time external environment changes. Therefore, this study proposes a louver that can change shapes by integrating the characteristics of horizontal and vertical louvers. The details are as follows.

First, as shown in Figure 3, the proposed louver has a hybrid structure that combines the characteristics of horizontal and vertical louvers; however, the shape can be changed by operating the system. As shown in Figure 3, the solar tracking-based movable louver (STML) system has four control points by combining the horizontal and vertical louvers. This enables modifications to the system's awning shape by adjusting the length of each control point. The movable part used for the four control points was developed based on the arms of the awning systems, which allow for length adjustments [42]. Spandex was applied to the awning material of the system. Spandex is a petroleum compound primarily composed of polyurethane. It is strong, flexible, and has excellent elasticity and resilience, stretching up to eight times its original length [43]. These characteristics prevent material deformation and damage caused by repeated contraction and expansion of the louver awning. Second, using solar tracking, the proposed louver system adjusts the shape of the awning to align with the sun. This process consists of tracking the sun and operating the louver arm. The system employs solar tracking sensors to minimize the influence of weather and adjacent buildings. For this purpose, illuminance sensors were installed at the end of each louver arm, and the illuminance value of a total of four sensors was monitored; the sensor with the highest illuminance value among them is located closest to the sun. The length of the louver arm, which shows the highest illuminance value, is sequentially increased to expand the area of the awning shade. Based on the mechanism described above, the proposed louver system can block direct sunlight entering the indoor space based on its morphological characteristics. It can also create a comfortable

lighting environment while saving energy by admitting external natural light into the room through diffusion and scattering. Furthermore, using multiple slats, it can partially address issues inherent to conventional louvers, such as obstructing the daylighting window view. However, unlike traditional louvers with multiple slats, the proposed system forms the awning shade at the end of the daylighting window. In this regard, the longer protruding length for efficient shading may be disadvantageous.

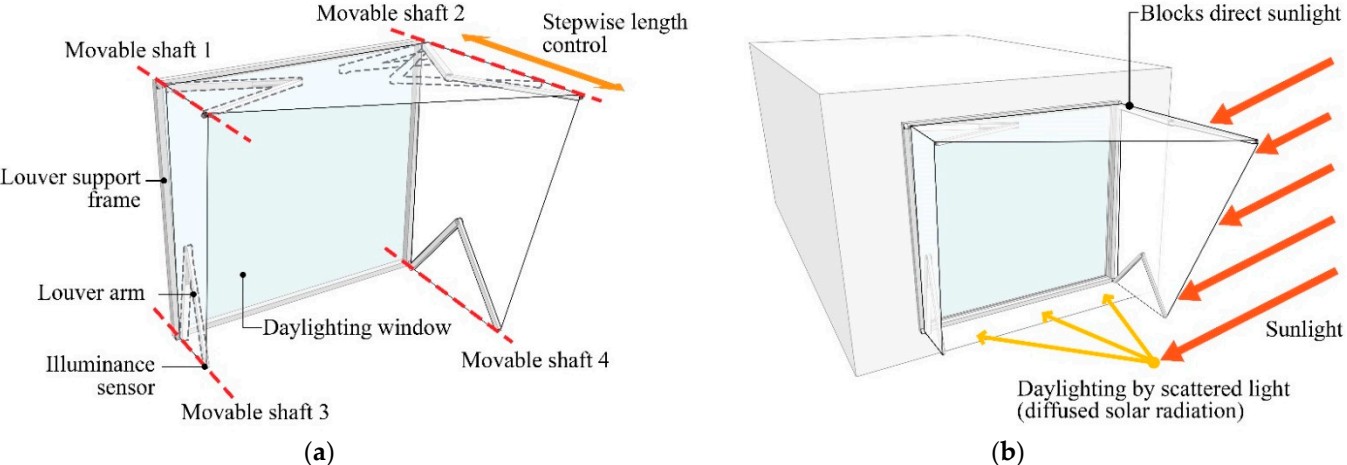

**Figure 3.** Structure and operation of solar tracking-based movable louver (STML) system. (**a**) Structure and operation principle. (**b**) Concept of daylighting and shading.

*2.2. Performance Evaluation Environment*

A full-scale test bed based on an artificial climate chamber capable of creating an artificial external environment was built to evaluate the performance of the proposed louver system. Because of the advantages of creating external environments (altitude and azimuth of the sun) and implementing the same environment for each case, the performance evaluation was conducted in an artificial environment. The details of the test bed and artificial environment chamber are as follows.

First, the dimensions of the internal space of the test bed were 4.9 m (W) × 6.6 m (D) × 2.5 m (H). The area of the daylighting window to install the louver was 1.9 × 1.7 m. Figure 4 shows the detailed specifications. However, the location of the daylighting window in the test bed was skewed to one side rather than in the center of the wall. Additionally, 10 illuminance sensors were placed in the indoor space. Based on the research result [44], the locations were adjusted so that the best distance for measuring indoor illumination was 4.4 m from the daylighting window. Given the height of the working surface, the illuminance sensors were positioned 0.85 m from the floor, with a temperature sensor installed in the center of the indoor space. Four LED-type lights were installed inside the test bed, each with 8-level dimming control. Additionally, illuminance sensors 2, 4, 7, and 9 were paired with lights 1, 2, 3, and 4, respectively, to control the lighting in the indoor space. Figure 5 displays the light distribution curve and the conical illuminance of the lighting. Furthermore, an air conditioner was installed in the test bed. It operated in conjunction with the temperature sensor located in the center of the indoor space and not with the temperature sensor embedded in the air conditioner.

Second, an artificial climate chamber was built to implement an artificial environment. The artificial climate chamber had an artificial solar irradiation apparatus and chamber thermostat to simulate the sun and control the chamber's temperature. The artificial solar irradiation apparatus included an artificial light source that allowed us to adjust the altitude and intensity of the sun. However, due to its mechanical limitations, this apparatus could only simulate azimuth angles between 120° and 230°. These aspects can be considered a limitation of this study. However, in terms of illuminance uniformity and time variation, this device is a Grade-A artificial solar irradiation apparatus according to

ASTM E927-85. As such, the results obtained from the experimental environment are valid and highly reliable.

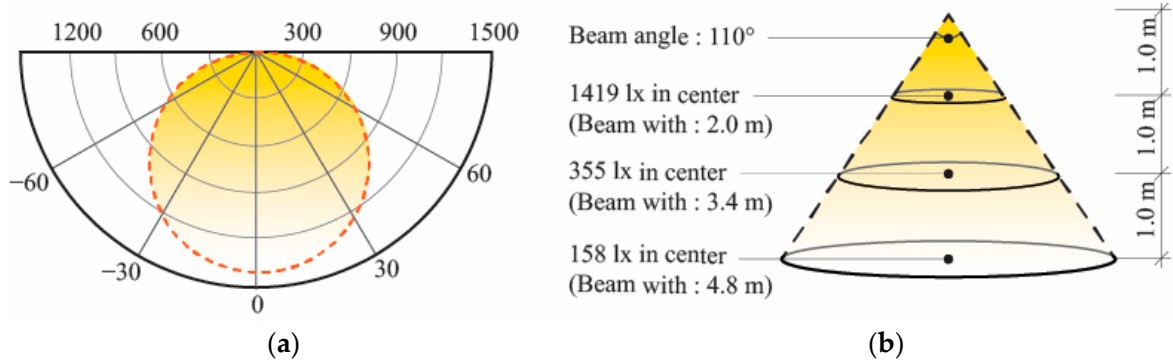

■ : Lighting (L−n : Light number)

◎ : Illuminance sensor (I−n : Illuminance number)

● : Temperature sensor

**Testbed**
· Size: 4.9 m (W) x 6.6 m (D) x 2.5 m (H)
· Reflexibility: Ceiling (86 %), Wall (46 %), Floor (25%)
· Wall material : Insulation panel (Thk 100mm)

**Artificial climate chamber**
**Chamber thermostat**
· Range : −20 ~ 50℃
· Precision : +1℃

**Daylight window**
· Size: 1.9 m (W) x 1.7 m (H)
· Type: pair glass 24 mm (6 mm + 12 mm + 6 mm)
· Transmissivity: 80 %
· Thermal transmittance : 2.83W/m²K(Summer), 2.69W/m²K(Winter)

**Artificial solar irradiation apparatus**
· Light soure with adjustable intensity
· Light soure with adjustable height/angle
· Precision of solar light radiation : Grade A
　(according to ASTM E927−85)

**Temperature sensor**
· Sensing element: NTC 10KΩ: AN Type
· Detection range: −40 ~ +90℃
· Precision: ±0.3℃

**Illuminance sensor**
· Type: Pair glass 24mm (6mm+12mm+6mm)
· Thermal transmittance: 2.83W/m²K(Summer), 2.69W/m²K(Winter)
· Transmissivity: 80%

**Lighting and dimming controller**
· 8 Level dimming (LED type) 4ea
· Dimension (mm): 600 x 600
· Dimming Range: 10 ~ 100 %
· Heating temperature: 35℃

**Air conditioner**
· Model : AP−SM302(EHP)
· Heating capacity : 13,200w
· Cooling capacity : 11,000w
· COP : 3.38℃ (Heating), 2.82℃ (Cooling)

**Energy monitoring system**
· Model: SPM−141
· Measurement capacity: Single phase (220 V, 1−50 A)
· Error rate: Within 2.0 %
· Measurement items: power, voltage, current (real−time, and accumulated amount)

**Figure 4.** Overview of the test bed.

Beam angle : 110°

1419 lx in center (Beam with : 2.0 m)

355 lx in center (Beam with : 3.4 m)

158 lx in center (Beam with : 4.8 m)

(**a**)　　　　　　　　　　　　　　　　　　(**b**)

**Figure 5.** The light distribution curve and conical illuminance of lighting. (**a**) Light distribution curve. (**b**) Conical illuminance.

### 2.3. Performance Evaluation Method

The performance of the proposed STML system in terms of saving building energy and improving the light environment was evaluated as follows.

First, as shown in Figure 6, a test specimen was fabricated for the performance evaluation. However, rather than applying the louver arm proposed in Section 2.1, a profile was made considering the range of motion of the louver. The shape of the awning was changed along the profile's rail. In addition to the ease of production, the test specimen was developed with a profile because it allows for precise control between performance evaluations. However, regardless of the awning shape, the shape of this specimen must protrude over a certain length, making it difficult to apply in real-world settings.

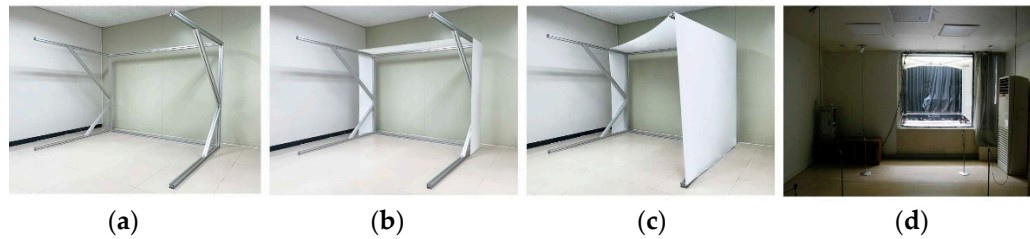

|     (a)     |     (b)     |     (c)     |     (d)     |

**Figure 6.** Test specimens for performance evaluation: (**a**) fabrication of moving parts (profile), (**b**) awing installation, (**c**) awning control, and (**d**) installation in the test bed.

Second, as shown in Table 3, five cases were configured to validate the effectiveness of the proposed STML system (Case 5). Case 1 was set as a daylighting window without a louver installed. Case 2 was a vertical louver with seven slats, and the width, depth, and height of each slat were set to 0.03, 0.25, and 1.65 m, respectively, based on a related study [28]. The interval between each slat was 0.29 m. Case 3 was a horizontal louver with six slats. Case 4 was an eggcrate louver with a structure that combined Cases 1 and 2. As shown in Figure 7, the slats in Cases 2 and 3 could be controlled to different angles (90°, −45°, 0°, 45°) relative to the daylighting window. In Case 4, the slats were fixed at an angle perpendicular to the window. The slats in Cases 1, 2, 3, and 4 were made of aluminum with 70% reflectance. Furthermore, the louver arms in Case 5 could be controlled in eight steps, ranging from a minimum of 0.2 m to a maximum of 1.6 m, as shown in Figure 8.

**Table 3.** Case settings for performance evaluation.

| Case | Louver Type | Operation | # of Slats | Slat Specifications | Slat Intervals | Slat Angle |
|------|-------------|-----------|------------|---------------------|----------------|------------|
| 1 |  |  |  | Louver not applied |  |  |
| 2 | Vertical | Fixed | 7 | 0.03 m (W) × 0.25 m (D) × 1.65 m (H) | 0.29 m | −90°, −45°, 0°, 45° |
| 3 | Horizontal | Fixed | 6 | 1.85 m (W) × 0.25 m (D) × 0.03 m (H) | 0.30 m | −90°, −45°, 0°, 45° |
| 4 | Eggcrate | Fixed | 13 | Combination of vertical and horizontal louvers | Vertical 0.24 m, Horizontal 0.23 m | 0 |
| 5 | Hybrid | Movable | 3 | 8-step width control of louver arm (Awning width: 0.2 m increments from 0.2 m to 1.6 m) | - | - |

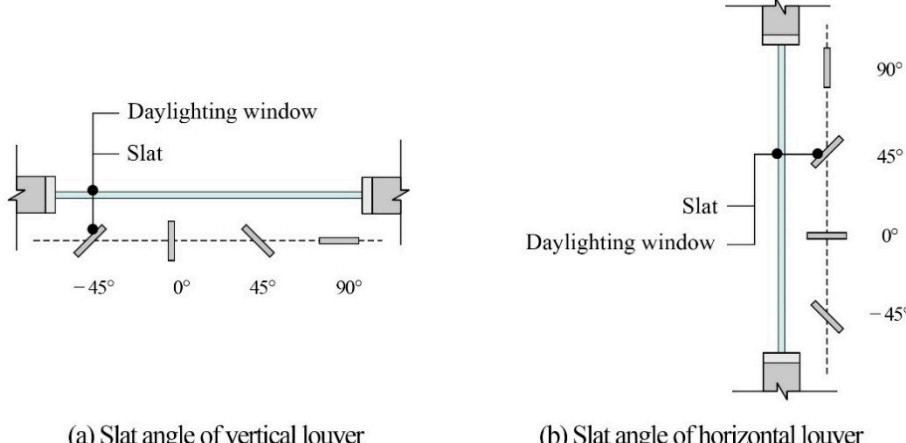

**Figure 7.** Slat angles of vertical and horizontal louvers: (**a**) Slat angle of vertical louver, (**b**) Slat angle of horizontal louver.

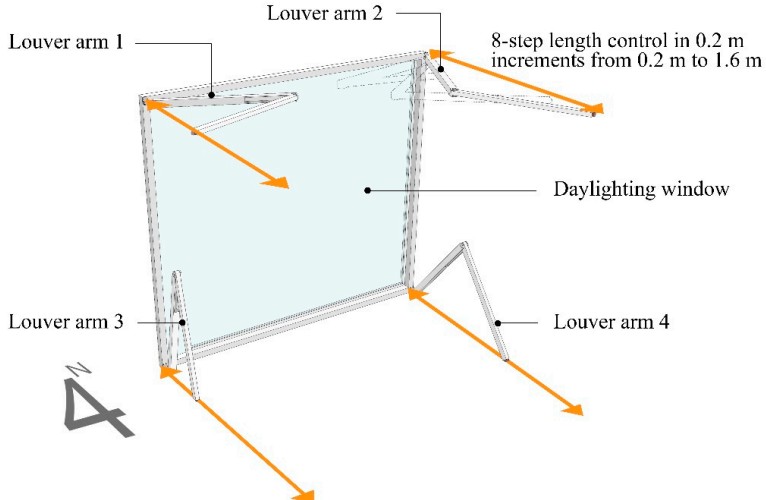

**Figure 8.** Louver arm control of the solar tracking-based movable louver (STML).

Third, this study derived the illuminance uniformity of each case and the lighting energy consumption to create a comfortable visual environment. Indoor uniformity was used as a performance evaluation indicator because it is related to the occupants' visual comfort and a quantitative indicator that directly or indirectly expresses the shape and amount of natural light flowing in from the external environment [45,46]. For example, because of the high solar altitude in the summer, natural light does not penetrate deep into the room, causing uniformity to decrease. Therefore, the flow of natural light must be blocked to improve uniformity. In this study, the uniformity ratio was derived based on the values of 10 illuminance sensors in the indoor space. Additionally, the illuminance uniformity was calculated as the ratio of the minimum illuminance to the maximum illuminance value of the indoor space. Lighting energy consumption was derived based on light dimming control to maintain optimal indoor illuminance. Light dimming control was performed only when the minimum values of illuminance sensors 2, 4, 7, and 9, which were paired with the lights, were less than 500 lx. The dimming control process was based on the following. Light dimming control increased the lighting's dimming levels sequentially from the light paired with the illuminance sensor, showing the minimum value among the illuminance sensors 2, 4, 7, and 9. Dimming control ended when the minimum value of the illuminance sensors 2, 4, 7, and 9 reached 500 lx. For example, if illuminance sensor no. 2 has the minimum value (300 lx), the dimming level of light no. 1, which is paired with illuminance sensor no. 2, is increased incrementally from level 1 to 8, with illuminance

sensors 2, 4, 7, and 9 monitored to determine whether they reach 500 lx. However, if the values of illuminance sensors 2, 4, 7, and 9 do not reach 500 lx despite increasing the dimming level of light no. 1 to level 8, dimming control is transferred to light no. 3, the closest light to light no. 1, with illuminance sensors 2, 4, 7, and 9 being checked whether they reach 500 lx. During this process, derivation of the lighting energy consumption is made based on the light dimming control when the values of the illuminance sensors 2, 4, 7, and 9 satisfy 500 lx. The optimal indoor illuminance was set to 500 lx based on the indoor illuminance standards in the US, Japan, Korea, and Europe [47–50].

Fourth, this study derived the amount of air conditioning required to maintain the optimal indoor temperature according to each case and specification. Based on a related study [51], the optimal temperatures for summer and winter were set at 26 °C and 20 °C, respectively. The air conditioner automatically maintained a constant temperature through a built-in function without any manual control. The accuracy of controlling the air conditioner was improved by pairing the temperature sensor in the center of the indoor space with the air conditioner rather than using the sensor embedded inside the air conditioner. The effectiveness of reducing the heating and cooling energy in each case was verified by monitoring the energy consumed for operating the heating and cooling equipment.

Fifth, the external environment setting for performance evaluation was restricted to South Korea (Seoul) due to the country having distinct seasonal characteristics. Table 4 shows the criteria for setting the environment based on the 30-year average climate data from the Korea Meteorological Administration [52]. The time range was 10:00 to 15:00. However, due to the mechanical characteristics of the artificial solar irradiation apparatus, the azimuth for each period differed from the actual environment. Additionally, the solar radiation for each period was set as the intensity of the lighting. These aspects are the limitations of this study.

**Table 4.** Solar altitude and external illuminance during summer, mid-season, and winter.

| Season | Meridian Altitude | Outdoor Temperature | External Illuminance, Azimuth, and Solar Radiation by Time | | | | |
|---|---|---|---|---|---|---|---|
| | | | 10:00–11:00 | 11:00–12:00 | 12:00–13:00 | 13:00–14:00 | 14:00–15:00 |
| Summer | 76.5 | 27.1 °C | 70,000 lx, 120°, 530 W/m² | 80,000 lx, 147°, 638 W/m² | 80,000 lx, 174°, 638 W/m² | 80,000 lx, 201°, 638 W/m² | 70,000 lx, 228°, 530 W/m² |
| Winter | 29.5 | −3.2 °C | 20,000 lx, 120°, 289 W/m² | 30,000 lx, 147°, 289 W/m² | 30,000 lx, 174°, 332 W/m² | 30,000 lx, 201°, 332 W/m² | 20,000 lx, 228°, 289 W/m² |

## 3. Performance Evaluation Results and Discussion

### 3.1. Performance Evaluation Results

The performance of the solar tracking movable louver system was evaluated using a test bed to validate its effectiveness. The results are as follows.

First, Figure 9 shows the results of an analysis of the indoor uniformity of Cases 1, 2, 3, and 4, which do not have a louver or have conventional types of louvers. This shows that installing louvers improves indoor uniformity. In Case 1, where no louvers were installed, the uniformity differed according to the sun's position for each period during the summer and winter. The uniformity was lower in the winter compared with the summer, resulting in a low comfort level for visual work. Cases 2 and 3, which correspond to vertical and horizontal louvers, exhibited differences in efficiency depending on the sun's position. When the sun was positioned to the southeast and southwest, the vertical louver outperformed the horizontal louver in terms of improving the light environment. However, even with the vertical louver, the uniformity decreased between 13:00 and 15:00. This is contrary to the fact that vertical louvers are advantageous when the sun is positioned to the southwest. The horizontal louver exhibited better performance when the sun was close to the south. As shown in Table 5, the vertical and horizontal louvers had different optimal

angles for improving the indoor light environment. The eggcrate louver was effective in improving indoor uniformity due to the combination of vertical and horizontal louvers. Based on optimal specifications, Cases 2, 3, and 5 improved indoor uniformity by 43.3%, 37.5%, and 49.7%, respectively, on average, compared with Case 1.

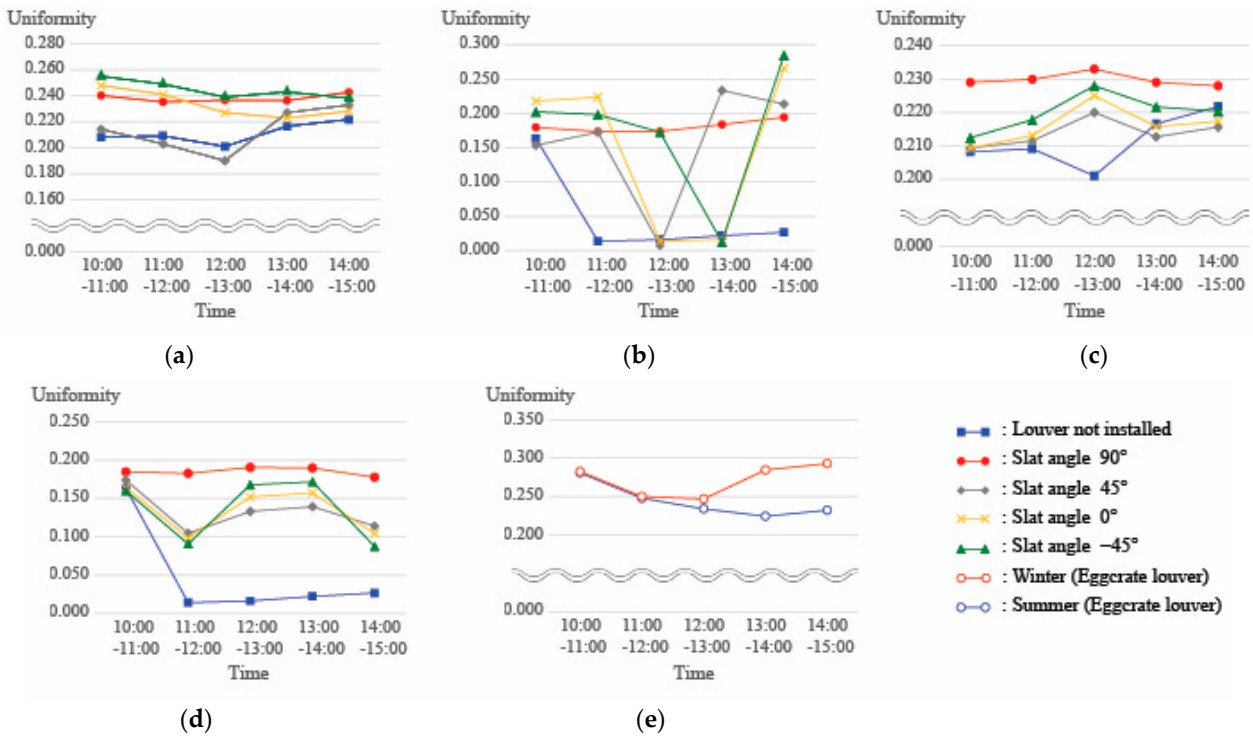

**Figure 9.** Results of analyzing the indoor uniformity according to Cases 1, 2, 3, and 4. (**a**) Vertical louver in summer. (**b**) Vertical louver in winter. (**c**) Horizontal Vertical louver in summer. (**d**) Horizontal Vertical louver in winter. (**e**) Eggerate louver.

**Table 5.** Optimal slat angles for each period in Cases 2 and 3 to improve indoor uniformity.

| Case | Optimal Slat Angle in Summer (Uniformity) | | | | | Optimal Slat Angle in Winter (Uniformity) | | | | |
|---|---|---|---|---|---|---|---|---|---|---|
| | 10:00– 11:00 | 11:00– 12:00 | 12:00– 13:00 | 13:00– 14:00 | 14:00– 15:00 | 10:00– 11:00 | 11:00– 12:00 | 12:00– 13:00 | 13:00– 14:00 | 14:00– 15:00 |
| 2 | −45 (0.255) | −45 (0.249) | −45 (0.239) | −45 (0.243) | 90 (0.243) | 0 (0.218) | 0 (0.224) | 90 (0.174) | 45 (0.234) | −45 (0.285) |
| 3 | 90 (0.229) | 90 (0.230) | 90 (0.233) | 90 (0.229) | 90 (0.228) | 90 (0.185) | 90 (0.183) | 90 (0.191) | 90 (0.190) | 90 (0.178) |

Second, Figures 10 and 11 show the indoor uniformity results based on the stepwise control of the proposed STML system (Case 5). According to the method presented in Section 2.1, the length of the louver arms in Case 5 increased sequentially from 10:00 to 13:00 in the order of louver arms 2, 4, 1, and 3, thereby increasing the shading area. Furthermore, from 13:00 to 15:00, the louver arms operated sequentially in the following order: 1, 3, 2, and 4. During the summer, Case 5 tended to improve indoor uniformity by increasing the shading efficiency through stepwise operation control and increasing the shading area. However, above a certain level, uniformity did not increase even when the shading area increased; rather, it decreased in uniformity in some sections. Due to the relatively low solar altitude in the winter, the uniformity increased as the shading area increased before 13:00. Based on the above, Table 6 shows the optimal specifications for improving uniformity in Case 5. According to these specifications, Case 5 improved indoor uniformity by 46.1%, 5.0%, and 13.9%, compared with Cases 1, 2, and 3, respectively.

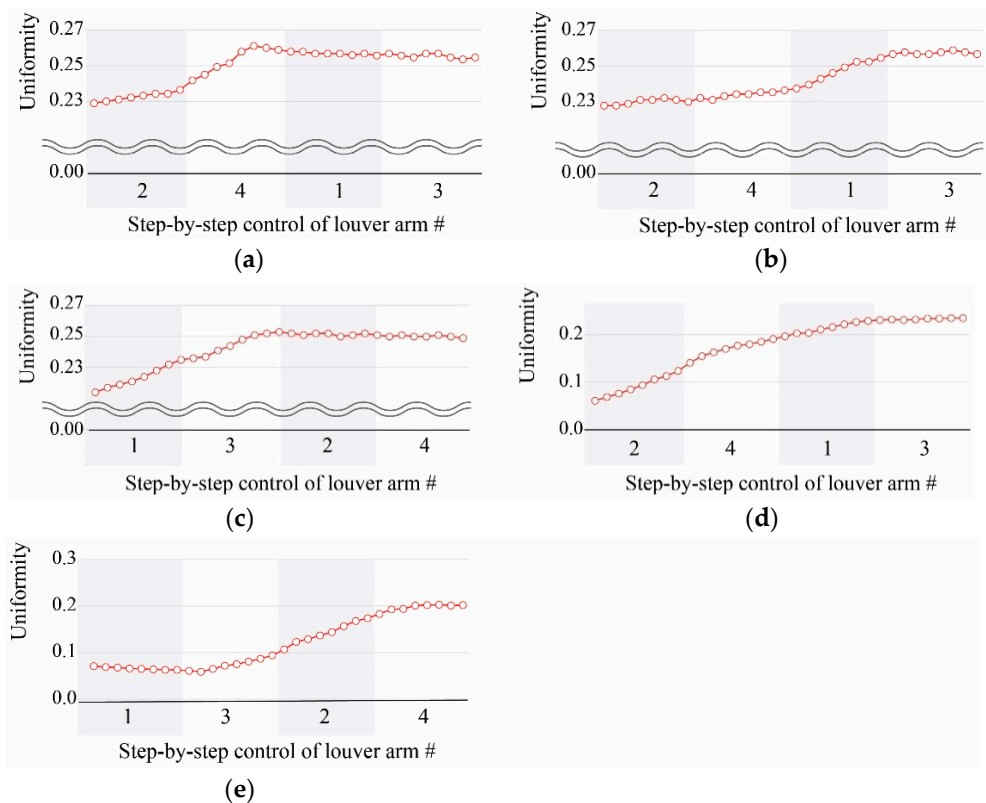

**Figure 10.** Uniformity by time during the summer according to operating Case 5. (**a**) From 10:00 to 11:00. (**b**) From 11:00 to 12:00. (**c**) From 12:00 to 13:00. (**d**) From 13:00 to 14:00. (**e**) From 14:00 to 15:00.

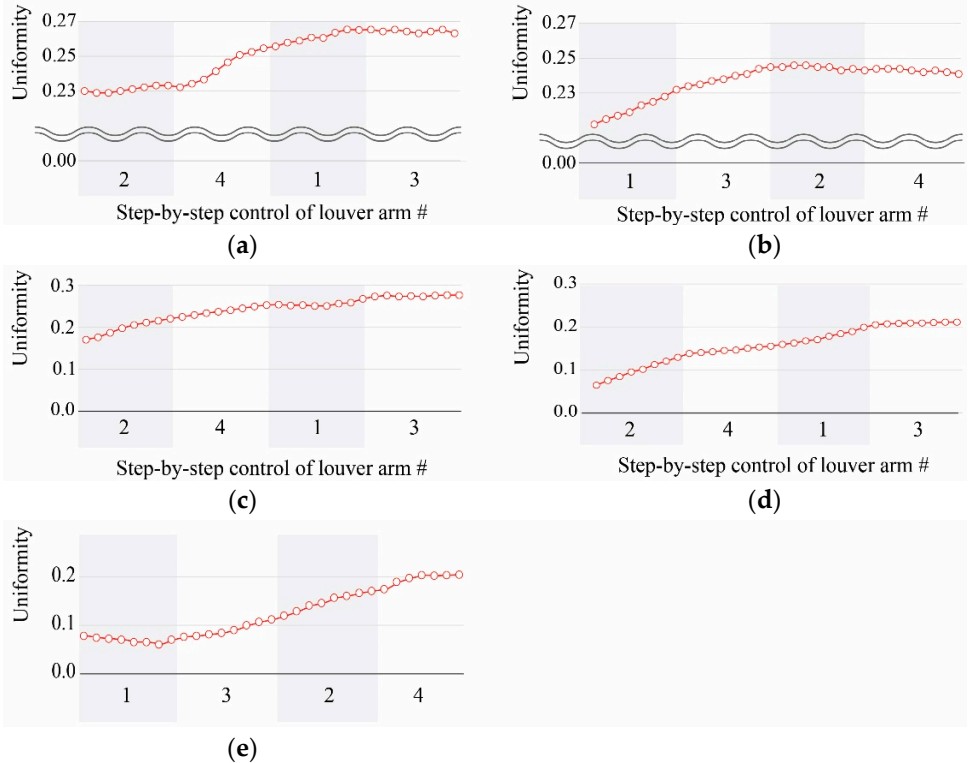

**Figure 11.** Uniformity by time during the winter according to operating Case 5. (**a**) From 10:00 to 11:00. (**b**) From 11:00 to 12:00. (**c**) From 12:00 to 13:00. (**d**) From 13:00 to 14:00. (**e**) From 14:00 to 15:00.

**Table 6.** Optimal specifications and uniformity for each period in Case 5 for improving indoor uniformity.

| Season | Optimal Control Level by Time [Louver Arm #(Control Level)]/Uniformity/Awning Shape | | | | |
|---|---|---|---|---|---|
| | 10:00–11:00 | 11:00–12:00 | 12:00–13:00 | 13:00–14:00 | 14:00–15:00 |
| Summer | 2(8) + 4(6) /0.261/ | 2(8) + 4(8) + 1(5) /0.261/ | 2(8) + 4(8) + 1(8) + 3(8)/0.253/ | 1(8) + 3(8) + 2(2) /0.246/ | 1(8) + 3(8) /0.253/ |
| Winter | 2(8) + 4(8) + 1(8) + 3(8)/0.273 | 2(8) + 4(8) + 1(8) + 3(8)/0.235 | 2(8) + 4(8) + 1(8) + 3(8)/0.211/ | 1(8) + 3(8) + 2(8) + 4(8)/0.202/ | 1(8) + 3(8) + 2(8) + 4(8)/0.205/ |

Third, Figures 12 and 13 depict the lighting and heating/cooling energy consumption levels required to maintain the optimal indoor illuminance and temperature in Cases 1, 2, 3, 4, and 5. All cases require separate energy levels to maintain the optimal indoor illuminance and temperature based on the environmental factors set for performance evaluation. During the summer, Cases 2, 3, 4, and 5 used more lighting energy than Case 1 because the louvers blocked the inflow of natural light from the outside; however, the heating and cooling energy consumption decreased due to solar radiation blockage. That is, Case 1 was effective in lowering building energy consumption during the winter by securing solar radiation and saving energy consumption for heating. These results indicate that control louvers must be controlled in various ways to efficiently save energy in response to the external environment. Tables 7 and 8 show the optimal specifications required to save energy in Cases 2, 3, and 5, demonstrating how the differences in the optimal specifications for improving uniformity differ.

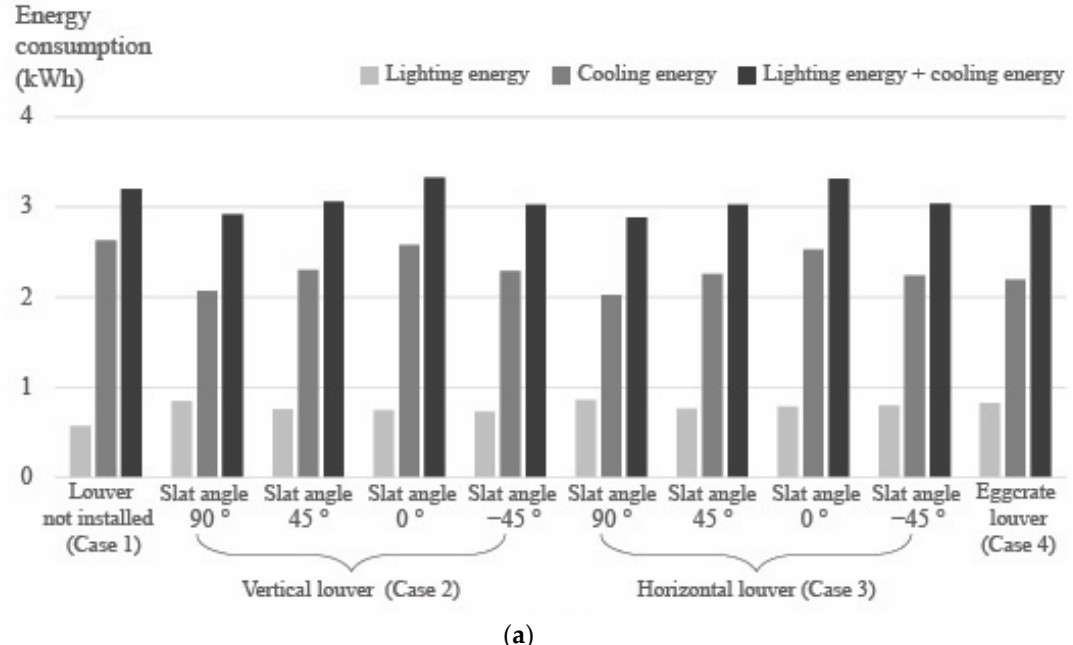

(**a**)

**Figure 12.** *Cont.*

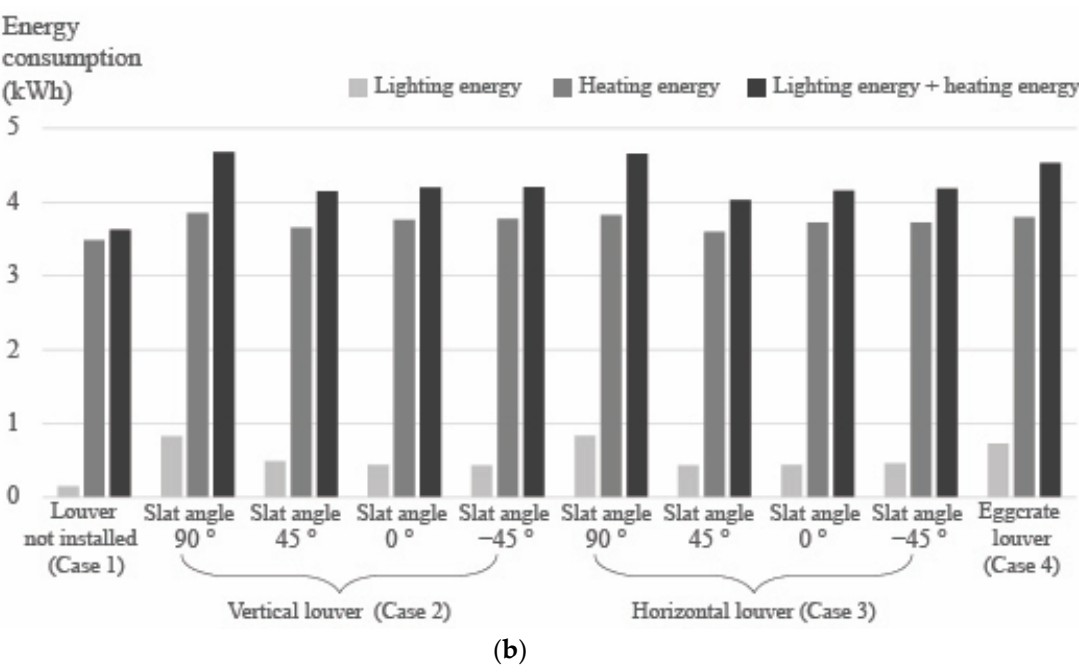

(**b**)

**Figure 12.** Analysis of energy consumption in Cases 1, 2, 3, and 4: (**a**) summer and (**b**) winter.

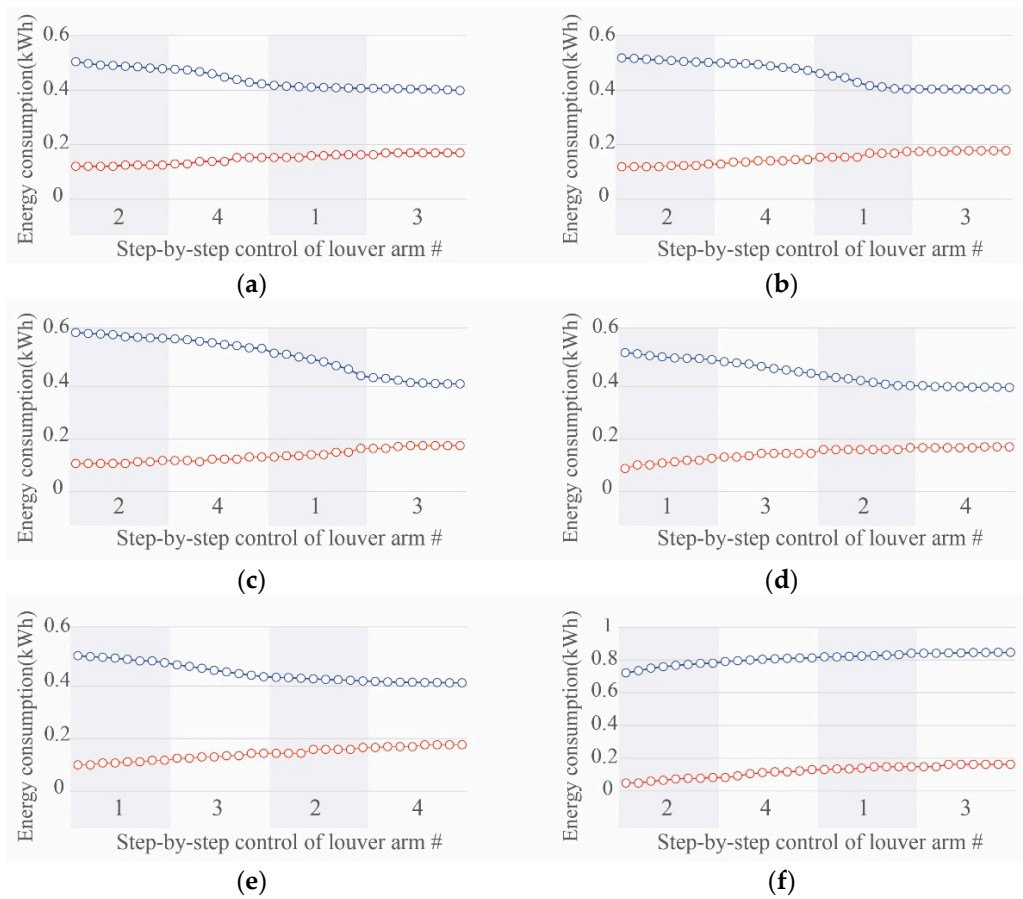

**Figure 13.** *Cont*.

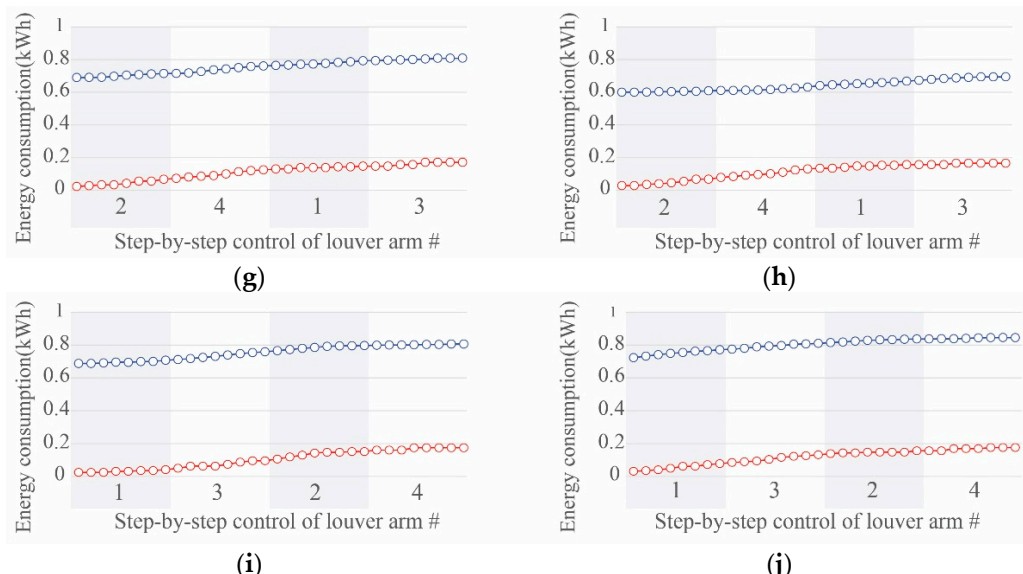

**Figure 13.** Analysis of lighting and cooling/heating energy consumption according to operating the proposed louver system. (**a**) From 10:00 to 11:00 in summer. (**b**) From 10:00 to 11:00 in winter. (**c**) From 11:00 to 12:00 in summer. (**d**) From 11:00 to 12:00 in winter. (**e**) From 12:00 to 13:00 in summer. (**f**) From 12:00 to 13:00 in winter. (**g**) From 13:00 to 14:00 in summer. (**h**) From 13:00 to 14:00 in winter. (**i**) From 14:00 to 15:00 in summer. (**j**) From 14:00 to 15:00 in winter.

**Table 7.** Optimal specifications and energy consumption by time in Case 5 for saving energy consumption.

| Case | Season | Optimal Specification for Each Period (Slat Angle) | | | | | Lighting and Heating/Cooling Energy Consumption (kWh) |
|---|---|---|---|---|---|---|---|
| | | 10:00–11:00 | 11:00–12:00 | 12:00–13:00 | 13:00–14:00 | 14:00–15:00 | |
| 2 | Summer | −45° | −45° | 90° | 45° | 45° | 7.184 |
| | Winter | 45° | 45° | 0° | −45° | −45° | |
| 3 | Summer | −45° | −45° | 90° | 0° | 0° | 7.094 |
| | Winter | 45° | 45° | 45° | 45° | 45° | |

**Table 8.** Optimal specifications for the proposed STML system for saving energy.

| Season | Optimal Louver Arm Specs for Each Period (Operation Level) | | | | | Lighting and Heating/Cooling Energy Consumption (kWh) |
|---|---|---|---|---|---|---|
| | 10:00–11:00 | 11:00–12:00 | 12:00–13:00 | 13:00–14:00 | 14:00–15:00 | |
| Summer | 2(8) + 4(8) + 1(3) | 2(8) + 4(8) + 1(6) | 2(8) + 4(8) + 1(3) + 2(7) | 2(8) + 4(8) + 1(7) | 2(8) + 4(8) + 1(3) | 6.555 |
| Winter | No operation | No operation | No operation | No operation | No operation | |

*3.2. Discussion*

In this study, a performance evaluation was conducted to prove the effectiveness of the STML.

First, vertical louvers were generally advantageous for southeast and southwest orientations, but performance results revealed that the uniformity decreased when the sun was positioned southwest. This is because the window of the test bed in this study was skewed to one side. External light should flow far away from the daylighting window to increase indoor uniformity: however, as shown in Figure 14, if the sun is positioned at the southwest after 13:00, the louver prevents natural light from penetrating deep into the indoor space, reducing uniformity. Particularly, this phenomenon may occur more frequently in the winter because the solar altitude is lower than in the summer. These

results indicate that the location of the daylighting window, as well as the characteristics of the indoor space, must be considered when designing louvers in the future.

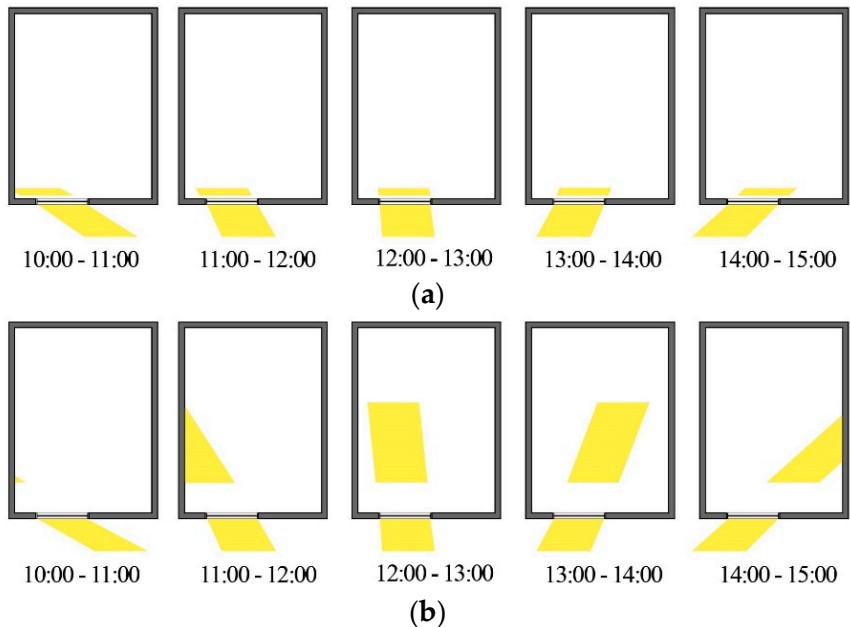

**Figure 14.** The inflow of natural light by time through the daylighting window in Case 1 (no louver): (**a**) summer and (**b**) winter.

Second, a performance evaluation was performed to measure improvements in the indoor lighting environment by deriving indoor uniformity based on the installation and operation methods of louvers. However, all cases necessitated energy consumption per the environment and variable settings established in this study. As such, energy savings were considered the sole criterion for deriving the optimal specifications for each case; this is because controlling the lighting modifies the distribution of indoor illuminance changes. However, different indoor illuminance standards are required depending on the nature of the indoor space. For example, if an indoor space requires low illuminance, the optimal louver specifications should be derived while considering decreases in uniformity and energy consumption.

Third, while conventional louver systems were effective in reducing cooling energy consumption by blocking solar radiation in the summer, blocking solar radiation increased heating energy consumption during the winter and was not suitable for energy savings. Therefore, it is preferable to remove the louver or keep it from blocking natural light from entering a room during the winter.

Fourth, the optimal specifications for each case were derived in terms of energy savings. Based on the results shown in Figure 15, the proposed louver system reduced energy consumption by 4.0%, 8.8%, 7.6%, and 13.1%, compared with Cases 1, 2, 3, and 4, respectively. This demonstrates the effectiveness of an STML. Furthermore, it reduced window view obstruction by installing the awning only outside the window rather than having slats crossing the daylighting window. However, the STML must protrude over a specific length for efficient shading.

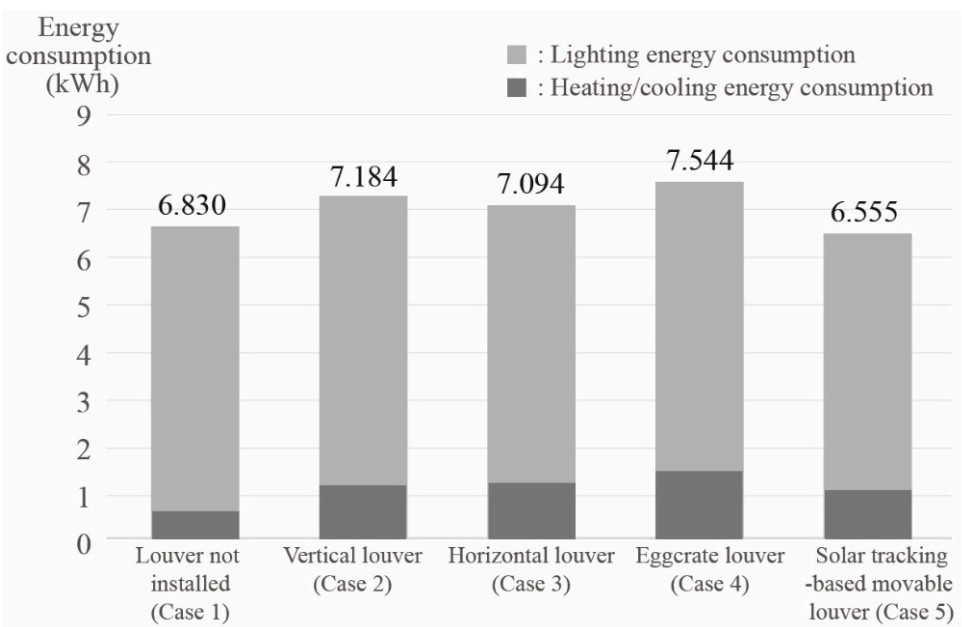

**Figure 15.** Lighting and heating/cooling energy consumption based on the optimal specifications of each case for saving energy.

## 4. Conclusions

This study proposed an STML that responds to various external environments efficiently, improves the indoor visual environment, and increases energy-saving efficiency. Furthermore, the effectiveness of the STML was validated through a performance evaluation. The main findings are as follows.

First, the proposed hybrid louver system has a structure that combines vertical and horizontal louver characteristics. The shape of the awning was changed by controlling each of the four movable shafts. Particularly, the shading and daylighting efficiency of the louver system were improved by incorporating solar tracking technology into sensors installed on each movable shaft. The proposed STML system also improved window view issues inherent in conventional louvers based on its structural characteristics. However, for efficient shading, the awning of this system must protrude beyond a certain length, which may be a disadvantage.

Second, a performance evaluation was conducted to measure improvement in the indoor uniformity of conventional horizontal and vertical louvers. The results showed that the closer the sun is to the south, the higher the awning efficiency of the vertical louver is when the horizontal louver faces southeast and southwest, which is beneficial for improving indoor uniformity. However, rather than focusing solely on the direction of the building, louver design must consider the window location and the characteristics of the indoor space. The reason for this is that the shading efficiency can change depending on the window location and the characteristics of the indoor space.

Third, the proposed louver system improved the indoor visual environment by operating similarly to conventional vertical and horizontal louvers. Particularly, the proposed STML system improved indoor uniformity by 5.0% and 13.9%, compared with the vertical and horizontal louvers, respectively. This proves the shading and daylighting effectiveness of the system.

Fourth, louvers are suitable for reducing building energy during the summer. However, they are not suitable for saving building energy in the winter because they increase heating energy by preventing solar radiation from entering an indoor space. In this respect, the proposed louver system was effective in saving energy even during winter by minimizing the awning shade. Additionally, the cases and variables set in this study require separate lighting energy consumption levels to maintain optimal indoor illuminance, implying that the optimal louver specifications can be derived by only considering energy



savings. As a result, based on the optimal specifications, the proposed louver system reduced energy consumption by 35.7–49.7% compared with previous louver technologies, proving its energy-saving effectiveness.

This study is significant because it proposes a new louver concept that can improve shading and daylighting efficiency by reviewing conventional louver technologies and validating their effectiveness through performance evaluation. However, in this study, performance evaluation was conducted under limited conditions based on an artificial environment. Another limitation of this study is the lack a technical review to assess the proposed technology's economic feasibility. Further research should improve the limitations of this study.

**Author Contributions:** Conceptualization, S.-y.J. and S.H.; Methodology, S.-y.J. and M.-G.L.; Writing—Original draft preparation, S.-y.J. and S.H.; Writing—Review and editing, S.-y.J. and H.L.; Supervision, H.L. All authors have read and agreed to the published version of the manuscript.

**Funding:** This work was supported by a National Research Foundation of Korea (NRF) grant funded by the Korean government (MSIT) (NRF-2020R1C1C1004704). This work was supported by the Energy Demand Management Core Technology Development of the Korea Institute of Energy Technology Evaluation and Planning (KETEP) who granted financial resources from the Ministry of Trade, Industry, and Energy, Republic of Korea (No. 20212020900380).

**Institutional Review Board Statement:** Not applicable.

**Informed Consent Statement:** Not applicable.

**Data Availability Statement:** No additional data are available.

**Conflicts of Interest:** The authors declare no conflict of interest.

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
