# Peer review of "Development of a Solar Tracking-Based Movable Louver System to Save Lighting Energy and Create a Comfortable Light Environment"

_buildings, doi:10.3390/buildings12112017_

Round 1
Reviewer 1 Report
The paper presents a concept for an optimal type of louvre for use in buildings and its control method. The concept is evaluated under test conditions and compared to conventional shading methods in a highly controlled indoor environment using illuminance uniformity and energy consumption of lighting and air-conditioning as the performance evaluation indicators.
Factors for indoor climate comfort such as air velocity, occupants' clothing, activity level, noise and air quality are not included in the experiments and the results are therefore limited to the two variables which are included. Further work should also evaluate users' perception of indoor climate comfort levels, as uniform lighting may not be ideal for many. The design of the louvre can be improved with time as it may not appeal to many architects' aesthetic sensibilities.
It is not clear what the energy measurements include - do the measurements for example include the extensive technology employed to run sensors, mechanical movements of the louvres, etc.?
However, the paper presents an interesting and innovative concept for three dimensional, flexible louvres; the methods are thorough and sound; and the results promising, although in the early stage of design development.
Author Response
According to the reviewers’ comments, the manuscript underwent major revisions. All the changes are marked in the resubmitted manuscript. We elaborate on how we did this, and how we respond to the reviewer’s comments in a seprate file attached.

Reviewer 2 Report
This paper presents authors’ work on a method employed to develop a solar tracking-based movable louver (STML) system as shading devices for controlling daylighting. This aims to reduce energy consumption of the louvers through the use of solar tracking sensors and validating the effectiveness of the systems through a full-scale test bed.
The following needs to be improved
1) Abstract:
· The statement in line 14 to 15 should be restructured to be more concise e.g. A louver can perhaps be among technical component that are considered for energy performance in a building.
· A good abstract will contain brief and succinct description of the methodology before the itemizing the findings
2) Introduction
· Line 38- add of in the quoted phrase “how energy-efficient of a building is”
· The writing style should be improved for effectively communicating research outputs. For instance, in line 40, the main functions of the building envelope is not daylighting and shading, of course they may among functional requirements.
· Line 58 to 59, I am not sure if the statement is complete.
· Good description of solar tracking technology
3) Method.
· Some other overarching design and geometry consideration for automatic louver like this were not fully explore in this method. For instance, velocity and pressure drop and how this was catered for in this method.
· Interesting performance evaluation, Well-done!
· It appears that the case 2,3,4 and 5 are now shading devices in summer, which prevent excessive solar gain, thereby saving cooling energy. However, this becomes problematic in winter period if these louver does not have an automatic sensor that control the awning so that optimum solar gain can be utilized, by sensing the ambient temperature and irradiance to opening these louvers. Any consideration for this?
In summary this is a fantastic work!
Author Response

(The authors gave the same response as above.)

Reviewer 3 Report
Dear Authors,the study is really interesting an I appreciate your work.
Anyway, to improve the contribution, the louver design has to be better explained in relation to the integration into the envelope (is suitable for retrofit or new construction? Is louver compatible with the existing openings? What adaptability does the system have?) Furthermore, it is not clear how the energy consumption data are derived. Does the simulation depend on in situ monitoring? What data are selected for the energy calculation: those of electricity consumption or even HVAC for the temperature management? Then, it is necessary to understand the impact on internal comfort through a synthesis of thermal and lighting data: comfort cannot be deduced from the level of brightness alone. I suggest to expand the introduction part by inserting details on the new technology presented and its applicability; also insert the data that are missing into the results of comfort and energy consumption. Congratulations and good luck!
Author Response

(The authors gave the same response as above.)

Round 2
Reviewer 3 Report
I confirm that the work is interesting and that there is a lot of potential for the product presented in the paper. However, I regret that the authors did not fully accept the reviewers' suggestions to further improve the impact of their already important study.
Either way, I consider the work worthy of publication